# How Frequently Benign Uterine Myomas Appear Suspicious for Sarcoma as Assessed by Transvaginal Ultrasound?

**DOI:** 10.3390/diagnostics13030501

**Published:** 2023-01-30

**Authors:** Nieves Cabezas, Ana López-Picazo, Patricia Diaz, Beatriz Valero, María José Rodriguez, Ana Redondo, Begoña Díaz-de la Noval, Maria Angela Pascual, Silvia Ajossa, Stefano Guerriero, Juan Luis Alcázar

**Affiliations:** 1Department of Obstetrics and Gynecology, University Hospital Virgen Macarena, 41009 Seville, Spain; 2Department of Obstetrics and Gynecology, Clinica Universidad de Navarra, 31008 Pamplona, Spain; 3Department Obstetrics and Gynecology, Hospital Fundación Calahorra, 26500 Calahorra, Spain; 4Department of Obstetrics, Gynecology, and Reproduction, Hospital Universitari Dexeus, 08028 Barcelona, Spain; 5Department of Obstetrics and Gynecology, Central University Hospital Asturias, 33011 Oviedo, Spain; 6Centro Integrato di Procreazione Medicalmente Assistita (PMA) e Diagnostica Ostetrico-Ginecologica, Azienda Ospedaliero Universitaria—Policlinico Duilio Casula, Monserrato, University of Cagliari, 09042 Cagliari, Italy

**Keywords:** uterus, myoma, sarcoma, ultrasound

## Abstract

Background: Uterine myomas may resemble uterine sarcomas in some cases. However, the rate of benign myomas appearing as sarcomas at an ultrasound examination is not known. The objective of this study is to determine the percentage of benign myomas that appear suspicious for uterine sarcoma on ultrasound examination. This is a prospective observational multicenter study (June 2019–December 2021) comprising a consecutive series of patients with histologically proven uterine myoma after hysterectomy or myomectomy who underwent transvaginal and/or transabdominal ultrasound prior to surgery. All ultrasound examinations were performed by expert examiners. MUSA criteria were used to describe the lesions (1). Suspicion of sarcoma was established when three or more sonographic features, described by Ludovisi et al. as “frequently seen in uterine sarcoma”, were present (2). These features are no visible myometrium, irregular cystic areas, non-uniform echogenicity, irregular contour, “cooked” appearance, and a Doppler color score of 3–4. In addition, the examiners had to classify the lesion as suspicious based on her/his impression, independent of the number of features present. Eight hundred and ten women were included. The median maximum diameter of the myomas was 58.7 mm (range: 10.0–263.0 mm). Three hundred and forty-nine (43.1%) of the patients had more than one myoma. Using the criterion of >3 suspicious features, 40 (4.9%) of the myomas had suspicious appearance. By subjective impression, the examiners considered 40 (4.9%) cases suspicious. The cases were not exactly the same. We conclude that approximately 5% of benign uterine myomas may exhibit sonographic suspicion of sarcoma. Although it is a small percentage, it is not negligible.

## 1. Introduction

The prevalence of uterine myoma is very high. It has been estimated that between 35 and 80% of women will have a fibroid throughout their lives [1]. Almost 75% of patients are asymptomatic. The main symptom is metrorrhagia, followed by abdominal pain or a space-occupying mass that causes digestive and/or urological symptoms [2]. Uterine myoma is diagnosed mainly in the group of 30–50 years, being rare before 20 years [2].

At ultrasound examination, myomas are typically round or ovoid lesions located in the myometrium [3]. They are characteristically hypoechoic lesions, sometimes isoechoic, and more rarely hyperechoic. The echogenicity is usually non-homogeneous, with typical edge shadows. There are usually no cystic areas in the mass. They usually present a perilesional vascular ring [4]. However, myomas can change their appearance and undergo transformations, also known as “degeneration” [3,4]. It has been reported that the sensitivity and specificity of ultrasound for the diagnosis of uterine myoma are very high [3].

Uterine sarcomas constitute about 3% of all uterine cancers [5]. Worldwide, it is estimated that sarcomas have an incidence of 0.5 to 3.3 per 100,000 women/year [6]. The incidence of occult sarcoma in hysterectomy pieces due to suspected fibroids is about 0.2% [6,7]. Particularly, the risk for women under 40 is almost non-existent [7]. The vast majority of patients are symptomatic, with metrorrhagia being the most common symptom [8].

The specific diagnosis of uterine sarcoma is difficult and represents a true clinical challenge [9]. The introduction of morcellation as a surgical technique for laparoscopic removal of uterine leiomyomas or the uterus itself has made this challenge more problematic, since the prognosis appears to worsen in women with occult uterine sarcoma and a morcellated uterus [10]. After some years of controversy on this subject, several scientific societies have now cleared up the issue [11,12]. However, the preoperative diagnosis of uterine sarcomas continues to be a topic of concern to many clinicians and surgeons.

The largest series reported to date that analyzes the sonographic characteristics most commonly present in uterine sarcomas is that reported by Ludovisi et al. [13]. In a retrospective series of 179 cases, these authors showed that uterine sarcomas used to appear as heterogeneous, irregular uterine masses that frequently presented cystic areas and had no shadows. About three-quarters of uterine sarcomas are moderately or intensely vascularized. According to these authors, only 6–8% of sarcoma cases presented an image that could be considered characteristic of a benign myoma.

However, how often a benign myoma can look like a sarcoma has not been well studied. Therefore, we aimed to assess the percentage of benign myomas that appear suspicious for uterine sarcoma on ultrasound examination.

## 2. Materials and Methods

### 2.1. Study Design

This is a prospective multicenter study performed at six European centers between January 2019 and December 2021. Institutional Review Board (IRB) approval was obtained by the principal investigator of each center. All data were centralized at one institution for analysis (IRB approval number 2021.131).

### 2.2. Patients

Eligible patients were women with a diagnosis of having at least one uterine myoma diagnosed by ultrasound and scheduled surgery.

Inclusion criteria were as follows:Premenopausal or postmenopausal women diagnosed as having at least one myoma.Preoperative transvaginal or transrectal ultrasound.Surgical removal of myomas because of symptoms (bleeding, pain, discomfort), including hysterectomy (open or laparoscopic) and myomectomy (open, laparoscopic or hysteroscopic).Definitive histologic diagnosis of benign uterine myoma.

Exclusion criteria were as follows:Incomplete ultrasound data.Definitive histologic diagnoses other than benign myoma (uterine sarcoma, smooth tumor of uterine muscle of uncertain malignant potential, or any other diagnosis).

### 2.3. Ultrasound Assessment

All women underwent transvaginal or transrectal (in case of *virgo intacta*) exams by an expert examiner at each center. All principal investigators from each center agreed on a scanning protocol before the study started. In case more than one myoma was present, data from the largest one was used for analysis. In cases of large myomas, transabdominal ultrasound was performed.

We used the Morphological Uterus Sonographic Assessment (MUSA) consensus for describing the uterine and myoma ultrasound findings [4]. The following ultrasound features were specifically assessed for each myoma: lesion size (largest diameter of the three diameters in each orthogonal plane), lesion contour (regular or irregular), visible normal myometrium (yes or no), lesion echogenicity (uniform or not uniform), irregular cystic areas (yes or no), and intralesional color Doppler score (1 = no flow, 2 = scanty flow, 3 = moderate flow, 4 = abundant flow). We focused on these five features, as they were the features most frequently exhibited by uterine sarcomas according to the Ludovisi et al. study [13] (Table 1) (Figure 1, Figure 2 and Figure 3). Although not considered suspicious for sarcoma, we also recorded the presence of calcification (yes or no) and the presence of shadows (no, fan-shape or internal).

Additionally, the examiner had to classify the lesion as suspicious for sarcoma or benign myoma according to her/his subjective impression, taking into account ultrasound and clinical data (patient’s age and symptoms).

We also recorded whether magnetic resonance was performed before surgery by indication of the referring physician.

### 2.4. Reference Standard

All the women who were ultimately included underwent surgery, and a definitive histologic diagnosis of benign myoma was obtained. As stated above, patients with other histologic diagnoses were excluded.

### 2.5. Statistical Analysis

Continuous data are presented as mean with standard deviation or median with interquartile range depending on normal distribution or not. The Kolmogorov–Smirnov test was used to assess normal distribution. Categorical data are presented as numbers and percentages. We just performed a descriptive analysis of the data. No inferential statistics were performed.

## 3. Results

During the study period, eight hundred and ten (or 810) women were ultimately included. Patients’ mean age was 44.8 years old (standard deviation: 10.1), ranging from 30 to 81 years old. Three hundred and forty-nine women (43.1%) had more than one myoma. The mean largest diameter of the myomas was 58.7 mm (standard deviation: 1.3), ranging from 10 to 263 mm.

At ultrasound assessment, in 63 cases (7.8%), the myometrium was not visible. Lesion echogenicity was uniform in 555 cases (68.5%) and not uniform in 255 cases (31.5%). Irregular cystic areas within the lesion appeared in 86 cases (10.6%). The contour of the myoma was regular in 774 cases (95.6%), whereas it was irregular in 36 cases (4.4%). Four hundred and thirty-nine myomas (54.2%) had color score 1, two hundred and eighty-two myomas (34.8%) had color score 2, sixty-five myomas (8.0%) had color score 3, and twenty-four myomas (4.0%) had color score 4. In 558 cases (68.9%), there were no shadows. However, 164 myomas (20.2%) exhibited fan-shape shadowing, and 88 myomas (10.9%) had internal shadows. Calcifications were present in 46 (5.7%) myomas.

Table 2 shows how many cases would be suspicious of sarcoma according to the number of ultrasound features present. According to the examiner’s subjective impression, in the whole series, forty myomas (4.9%) were considered suspicious of sarcoma. This figure is almost identical when three or more features, according to Ludovisi’s criteria, are present (see Table 2).

The number of cases suspicious of sarcoma according to the examiner’s subjective impression was similar in all centers except one, where this figure was higher (Table 3).

Table 4 shows the distribution and number of Ludovisi’s criteria present and the examiner’s subjective impression.

Overall, MRI was performed in 67 cases (8.3%). Surprisingly, in most cases, the referring physician requested an MRI when there was no suspicion based on the examiner’s subjective impression (50 cases, 74.6%).

## 4. Discussion

As stated above, uterine myomas constitute the most frequent pelvic tumor in women. Risk factors for uterine myomas are well established and include African American ethnicity, the late onset of menopause, positive family history, a long interval since last delivery, hypertension, and consumption of dietary supplements and soy milk. Factors that reduce the risk of uterine myomas include multiple births, the use of oral contraceptives or depot medroxyprogesterone acetate, and smoking in women with a low body mass index [14,15]. On the other hand, uterine sarcomas are rare tumors with poor prognoses.

The differential diagnosis of myomas and sarcomas remains a challenge. Clinically, patients with uterine myomas or uterine sarcomas may exhibit similar symptoms, such as uterine bleeding, palpable pelvic mass, and pelvic pain. A recent retrospective case-control study involving 272 women with myomas and 68 women with sarcoma (43 leiomyosarcomas, 23 endometrial stromal sarcomas, and 2 undifferentiated sarcomas) observed that the presence of postmenopausal bleeding was the main factor associated with sarcoma (odds ratio: 35.3) [16]. Some attempts have been made to develop preoperative risk scores based on clinical data, such as the patient’s age, menopausal status, and bleeding symptoms, and tumor size [17,18]. The performance of these scores needs to be validated.

On the other hand, the traditionally rapid growth of a supposed myoma has been considered a factor in suspecting that lesion could be a uterine sarcoma. However, the prevalence of sarcoma in those lesions with “rapid growth” is very low (0.23%) [19]. In fact, it has been described that the occurrence of rapid growth in uterine sarcoma is lower than in uterine myoma [18]. In addition, the definition of “rapid growth” is unclear, and different authors used different definitions [17]. Parker and colleagues defined rapid growth as a gain of six weeks or more in gestational size during a year or less [19]. However, uterine myomas may duplicate the volume in one year [20].

Notwithstanding, imaging techniques are considered essential for discriminating uterine myomas from sarcomas [21]. The ultrasound features of uterine myomas have been well described in the literature. Typical uterine myomas used to appear as solid, well-defined, ovoid, or round lesions within the myometrium that were hypo or isoechoic when compared to the surrounding myometrium. Sometimes they may exhibit inhomogeneous echostructure. The presence of edge or fan-shape shadowing is not rare [3,4,14]. Classically, typical myoma vascularization, as assessed by a color Doppler ultrasound, has been described as having a perilesional or circumferential flow [3,4,14]. It has been reported that the sensitivity and specificity of transvaginal ultrasound for diagnosing uterine myoma are 99% and 91%, respectively [22].

However, uterine sarcomas remain one of the most difficult lesions to diagnose preoperatively when using ultrasound [23,24]. Many of the studies that analyze the sonographic characteristics of histologically confirmed uterine sarcomas report small series and do not distinguish between leiomyosarcomas and endometrial stromal sarcomas. Traditionally, the characteristic ultrasound image of uterine sarcoma is considered to be that of a single large mass of heterogeneous echogenicity that distorts the uterine contour, without acoustic shadows, with cystic areas and abundant vascularization, or a heterogeneous endometrial mass with abundant vascularization that penetrates the myometrium [24].

The use of pulsed Doppler and velocimetric indices, such as the resistance index or pulsatility index, was advocated for discriminating benign myomas from sarcomas. Kurjak et al. reported that uterine artery and myometrial vessels showed a significantly lower resistance index in cases of leiomyosarcoma as compared with normal uteri and uteri with benign leiomyomas [25]. However, the series for uterine sarcomas was small (n = 10). Hata et al. reported that the resistance index from uterine vessels was not significantly different in cases of leiomyosarcomas (n = 5) as compared with cases of benign uterine leiomyomas [26]. However, these authors found that peak systolic velocity was significantly higher in the case of leiomyosarcomas. A cut-off of >41 cm/sec showed a sensitivity of 80% and a specificity of 98% [27]. However, other authors did not find significant differences in pulsed Doppler velocimetric indices between LMS and benign myomas [28,29]. Due to these controversial results, the use of pulsed Doppler is not currently recommended. Exacoustos et al. described that tumor vascularization using color Doppler mapping was significantly higher in the case of uterine sarcoma as compared with benign myomas [30]. A recent study reporting data on 70 hypervascularized presumed benign uterine lesions showed that 93% of these lesions were actually benign lesions (32 typical leiomyomas, 29 leiomyoma variants, and 4 adenomyomas) and 7% were malignant lesions (two uterine endometrial sarcomas, one leiomyosarcoma, one neuroendocrine tumor, and one uterine smooth muscle tumor of uncertain malignant potential) [31]. All malignant lesions occurred in women older than 40 years old. Sixty percent of malignant lesions had irregular borders, compared to 1.5% of benign lesions. These data show that in the case of hypervascularized uterine lesions, they are most likely benign.

The largest series reported to date that analyzes which sonographic characteristics are most commonly present in uterine sarcomas is that reported by Ludovisi et al. [13]. In this study, nineteen expert sonographers analyzed retrospectively the images of 116 cases of leiomyosarcomas, 48 of endometrial stromal sarcomas, and 31 of undifferentiated endometrial sarcomas. They observed that, in the case of leiomyosarcomas, 94% are heterogeneous, 63% of the lesions have an irregular contour, 58% present cystic areas, and 71% have no shadows. As for Doppler studies, 73% of the leiomyosarcomas were moderately or intensely vascularized. The authors found that endometrial stromal sarcomas and undifferentiated endometrial sarcomas exhibited some differences regarding leiomyosarcomas. Endometrial stromal sarcomas and undifferentiated endometrial sarcomas had smaller mean sizes (68 mm and 70 mm, respectively, versus 106 mm for leiomyosarcomas). Visible normal endometrium and regular tumor borders were more frequently present in endometrial stromal sarcoma. Endometrial stromal sarcomas were less vascularized than the other sarcomas. Undifferentiated endometrial sarcomas had the highest rate of absent shadowing, irregular tumor borders, and hemorrhagic or ground-glass echogenicity within cyst areas when these areas were present.

According to these authors, about 20% of sarcomas were classified as benign by the original ultrasound report. However, after a consensus expert review of the images, only 6–8% of all uterine sarcomas presented an image that could be considered characteristic of a benign myoma. Endometrial stromal sarcoma was the type of sarcoma most often misclassified as benign on ultrasound, while undifferentiated sarcoma was the one least often misclassified as benign. The main limitation of this study is the retrospective design and the lack of analysis and comparison with benign myomas.

However, it is well known that benign myomas can suffer degenerative changes such as hyaline, myxoid, or cystic degeneration, dystrophic calcification, and red degeneration, the hyaline degeneration being the most frequent one [14]. These degenerative changes mostly affect the echogenicity of the lesion. Thus, echogenicity varies according to the different components in its context: muscle cells, fibrous stroma, calcification, and lipomatous or hyaline degeneration. When fatty tissue is overrepresented, myomas may appear hyperechoic; calcification or hyperechoic capsule, caused by deposition of calcium, is more frequent in postmenopausal women. When acute necrosis of the myoma occurs, complex heterogeneous internal structures or internal cystic areas frequently appear at ultrasound examination [14]. These changes produce an atypical appearance of the myomas and can be confusing, leading to an erroneous diagnosis of uterine sarcoma. [14].

We do think that our study can be clinically relevant because, to the best of our knowledge, we provide for the first time how frequently myomas can appear as sarcomas in a large series of myomas. This is the main strength of our study. According to our data, about 5% of myomas can mimic sarcoma. This figure is almost identical when expert examiners provide a diagnosis of suspicion based on her/his subjective impressions and when at least three ultrasound features reported by Ludovisi et al. [13], as frequently exhibited by sarcomas, are present at an ultrasound examination. This is interesting since these features could be used for a more objective assessment.

Another potential clinical impact of our study is related to the number of MRI assessments in cases of doubtful findings in the ultrasound assessment of reportedly uterine myomas. MRI is considered a good imaging technique for differentiating benign myomas from malignant sarcomas [32,33]. Typical findings for uterine leiomyosarcomas in MRI are high signals in T2WI, abnormal signals in DWI, high signals in T1WI, and ill-defined tumor mass borders [33]. In the case of low-grade endometrial stromal sarcomas, in MRI images, the mass exists from the uterine cavity to the myometrium, with a typical presentation in T1WI as low signals and T2WI as heterogeneous high signals. In contrast, in enhanced images, these lesions show moderate and heterogeneous contrast effects. The well-known characteristics of low-grade endometrial stromal sarcoma findings in MRI are the “worm-like” findings suggesting that this lesion is penetrating the normal myometrium while interposing itself intratumorally, and images that show worm-like interstitial and extrauterine extensions of multinodular mass. Regarding high-grade endometrial stromal sarcoma, the lesion used to occupy the uterine cavity and grow, with a presentation of heterogeneous signals for both T1WI and T2WI. These are large masses that may develop into hemorrhagic necrosis and infiltration. In many cases, HGESS shows heterogeneous contrast effects that are equivalent to or stronger than those of peripheral normal myometrium [33].

According to our data, in about 5% of all myomas, MRI could be advisable due to misleading ultrasound findings. However, we observed in our study that most patients submitted to MRI by the referring physician were cases in which suspicion was not raised either by the ultrasound examiner or by the presence of ultrasound features suspicious of sarcoma. This fact opens the question of reporting findings. Probably, in our study, the communication between the referring physician and ultrasound examiner failed. In our opinion, efforts must be made to improve this communication. Furthermore, our study is a multicenter study with six participating centers. We observed that results were similar in all centers. This gives an idea that our findings could be generalizable.

Our study has limitations. We did not consider clinical factors when assessing ultrasound features. All examiners were expert examiners, well trained with knowledge about how to describe ultrasound findings according to the MUSA consensus [4] and Ludovisi et al. study findings [13]. Probably, in less experienced hands, the percentage of suspicious cases would be higher. Additionally, we did not compare the ultrasound findings of myomas and sarcomas. Actually, this was beyond our goal in the present study. However, this comparison in large series is needed to accurately determine the diagnostic performance of ultrasound for discriminating between both entities. Furthermore, we did not include sarcomas in this study; therefore, no comparison between myomas and sarcomas could be performed.

## 5. Conclusions

In conclusion, we have observed that approximately 5% of myomas can exhibit features suggestive of sarcoma at an ultrasound examination. This information can be useful for sonographers and clinicians.

## Figures and Tables

**Figure 1 diagnostics-13-00501-f001:**
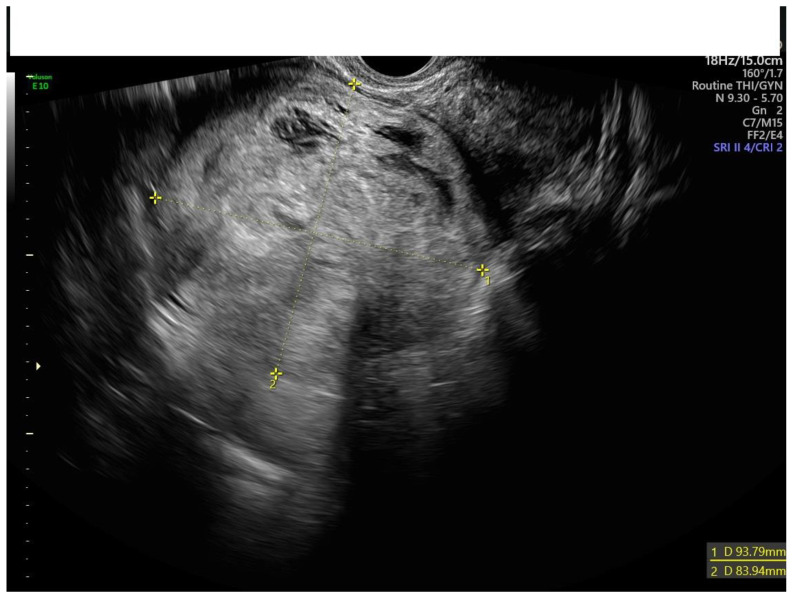
Transvaginal ultrasound of a myoma with no uniform echogenicity.

**Figure 2 diagnostics-13-00501-f002:**
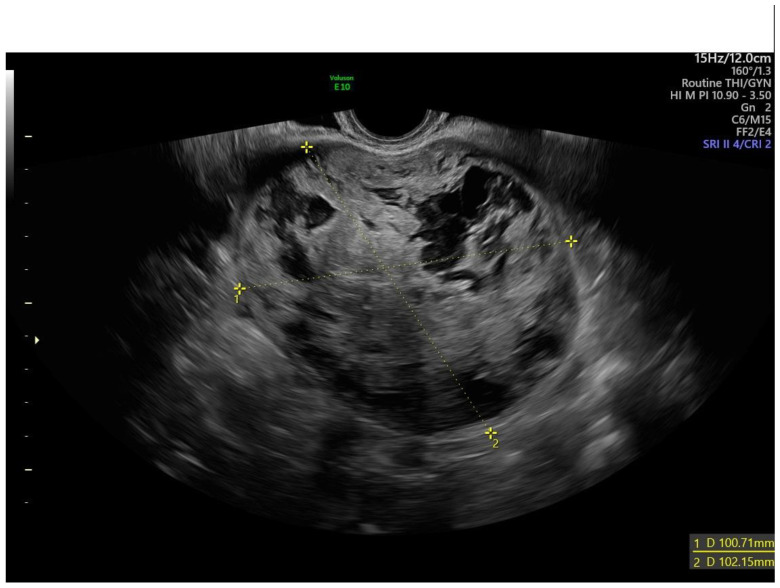
Transvaginal ultrasound of a myoma with irregular cystic areas.

**Figure 3 diagnostics-13-00501-f003:**
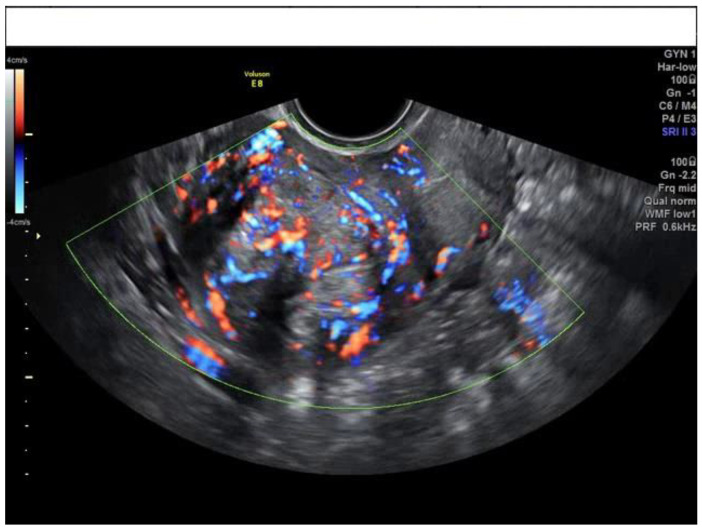
Transvaginal ultrasound of a myoma with an intralesional vascular score of 4.

**Table 1 diagnostics-13-00501-t001:** Ultrasound features more frequently present in uterine sarcomas *.

Ultrasound Feature	Frequency in Uterine Sarcomas
Irregular contour	52.8%
No visible myometrium	24.6%
Not uniform cchogenicity	77.4%
Irregular cystic areas	44.6%
Intralesional color score 3–4	67.9%

* Adapted from [13].

**Table 2 diagnostics-13-00501-t002:** Number of suspicious ultrasound features present in myomas in this series.

Ultrasound Suspicious Features	N (%)
None	478 (59.0)
One	203 (25.0)
Two	89 (11.0)
Three	24 (3.0)
Four	15 (1.9)
Five	1 (0.1)

**Table 3 diagnostics-13-00501-t003:** Distribution of suspicious cases according to examiner’s subjective impression in each participating center.

Suspicious	Center A	Center B	Center C	Center D	Center E	Center F
No	233 (95.1%)	189 (95.0%)	107 (96.4%)	125 (97.7%)	37 (94.9%)	79 (89.8%)
Yes	12 (4.9%)	10 (5.0%)	4 (3.6%)	3 (2.3%)	2 (5.1%)	9 (10.2%)
Total	245	199	111	128	39	88

**Table 4 diagnostics-13-00501-t004:** Distribution of a number of suspicious ultrasound features and examiner’s subjective impression.

	Suspicious Examiner’s Impression	
Number of Suspicious Ultrasound Features *	No	Yes
None	478 (100.0%)	0
One	198 (97.5%)	5 (2.5%)
Two	76 (85.4%)	13 (14.6%)
Three	15 (64.0%)	9 (36.0%)
Four	3 (20.0%)	12 (80.0%)
Five	0	1 (100.0%)

* According to data from [13].

## Data Availability

Data are available upon reasonable request.

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
