# Peer review of "How Frequently Benign Uterine Myomas Appear Suspicious for Sarcoma as Assessed by Transvaginal Ultrasound?"

_diagnostics, 2023, doi:10.3390/diagnostics13030501_

Round 1

Reviewer 1 Report

Generally I accept paper in the form as it is. Very interesting study with high impact for our practical clinical life. However small comments: line 56: what means 3,3% per 100000 women a year ?  Please claryfy that . 

line 153 - what means "en el abstract dice median cual es ? "

Author Response

  1. Question: Generally I accept paper in the form as it is. Very interesting study with high impact for our practical clinical life. However small comments: line 56: what means 3,3% per 100000 women a year ?  Please claryfy that.
    1. Answer: Thanks for this comment. Sorry for this mistake. Text amended. See line 58
  2. Question: line 153 - what means "en el abstract dice median cual es ? "
    1. Answer: Sorry for this mistake. Text amended. Text deleted.

Reviewer 2 Report

Well prepared work, with minimal need of changes (some mistypes)

The design of the work is very good.

The only part of the work that should be improved is the real diagnosis of sarcomas: either cases should be compared with negative and positive USG findings, or should be stated that this comparison has not been performed.

To state after this study that the percentage of bening/malignant tumors are this and this is not relevant. It sounds that dignity of the tumor can be determined by the USG findings.

Author Response

  1. Question: Well prepared work, with minimal need of changes (some mistypes). The design of the work is very good.
    1. Answer: thanks for this comment. We appreciate
  2. Question: The only part of the work that should be improved is the real diagnosis of sarcomas: either cases should be compared with negative and positive USG findings, or should be stated that this comparison has not been performed.
    1. Answer: We agree with the reviewer. We added a sentence in the Discussion. See line 329
  3. Question: To state after this study that the percentage of benign/malignant tumors are this and this is not relevant. It sounds that dignity of the tumor can be determined by the USG findings.
    1. Answer: We are sorry. We do not understand the question or whether the reviewer is questioning or just commenting. No change made.